# Forensic Diagnosis of Freshwater or Saltwater Drowning Using the Marker Aquaporin 5: An Immunohistochemical Study

**DOI:** 10.3390/medicina58101458

**Published:** 2022-10-15

**Authors:** Paolo Frisoni, Luca Diani, Stefania De Simone, Maria Antonella Bosco, Luigi Cipolloni, Margherita Neri

**Affiliations:** 1Unit of Legal Medicine, Azienda USL di Ferrara, Via Arturo Cassoli 30, 44121 Ferrara, Italy; 2Department of Biomedical, Metabolic and Neural Sciences, Institute of Legal Medicine, University of Modena and Reggio Emilia, Via del Pozzo 71, 41124 Modena, Italy; 3Department of Clinical and Experimental Medicine, Section of Legal Medicine, University of Foggia, 71122 Foggia, Italy; 4Department of Medical Sciences, Section of Legal Medicine University of Ferrara, Via Fossato di Mortara 70, 44121 Ferrara, Italy

**Keywords:** aquaporin-5, drowning, freshwater, seawater, forensic investigation, immunohistochemistry

## Abstract

*Background and Objectives*: Aquaporins are a family of water channel proteins. In this study, the renal and intrapulmonary expression of aquaporin-5 (AQP5) was examined in forensic autopsy cases to evaluate it as a drowning marker and to differentiate between freshwater drowning and saltwater drowning. *Materials and Methods*: Cases were classified into three groups: freshwater drowning (FWD), saltwater drowning (SWD), and controls (CTR). Samples were obtained from forensic autopsies at less than 72 h postmortem (15 FWD cases, 15 SWD cases, and 17 other cases) and were subjected to histological and immunohistochemical investigations. *Results*: In FWD group, intrapulmonary AQP5 expression was significantly suppressed compared with SWD and CTR; there was no significant difference in AQP5 expression among the other two groups. The same differences in expression were also observed in the kidney. *Conclusions*: These observations suggest that AQP5 expression in alveolar cells was suppressed by hypotonic water to prevent hemodilution. Moreover, it is possible to hypothesize that in the kidney, with the appearance of hypo-osmotic plasma, AQP5 is hypo-expressed, as a vital reaction, to regulate the renal reabsorption of water. In conclusion, the analysis of renal and intrapulmonary AQP5 expression would be forensically useful for differentiation between FWD and SWD, or between FWD and death due to other causes.

## 1. Introduction

Drowning represents the second leading cause of death from unintentional injury, after road traffic injuries [1]. According to the World Health Organization, 0.7% of all deaths worldwide each year are due to unintentional drowning. Drowning still represents a quite common suicidal method in many countries too [2]. The determination of the cause of death by drowning can be difficult since there are no specific pathognomonic features in post-mortem examination [3]. Consequently, the diagnosis of the cause and the establishment of the manner of death in submersion cases is still a challenge for forensic pathologists, as it often remains a diagnosis of exclusion mostly grounded on information from the recovery scene, the medical history, or witness reports. Gross autopsy findings, such as frothy fluid in the air passages (mouth, nostrils, etc.), lung distension and over-inflation, watery fluid in the stomach, and signs of asphyxia are not specific and do not prove that death resulted from drowning, since they are not peculiar to this mode of death. Furthermore, gross findings, namely frothy fluid in the airways or over-inflation of the lungs, can disappear by the time of the postmortem investigation. The most important and practically characteristic histological signs are acute water lung emphysema in drowning in freshwater, and important edema in drowning in saltwater. On the other hand, many attempts have been made to find histological hallmarks of drowning; however, no histological finding may be considered reliable to definitively assess the post-mortem diagnosis of drowning. Conclusively, macroscopic, and microscopic pathological findings are non-specific and, taken together, they may merely suggest a death caused by drowning [3,4].

Over the years, further different tests have been proposed as potential tools in determining drowning; in particular, the detection of diatoms in internal organs [5,6,7,8] has been considered as the “golden standard” for the diagnosis of drowning [9]. However, the real usefulness of the diatom test is still questioned for several reasons (i.e., lack of specificity, labor-intensive and time-consuming method, potential ante/post-mortem contamination, etc.). A further question that may be crucial in forensic practice is differentiating the drowning medium (freshwater—FWD, and saltwater—SWD, drowning).

This is a very subtle diagnosis since these two processes, despite having different pathophysiology, share a common pathway in brain and organ hypoxia [10]. The differential diagnosis between FWD and SWD is difficult, although from the pathophysiological point of view the mechanisms of death are very different because, in cases of individuals who died from FWD, the volume of circulating blood increases causes hypervolemia, marked hemodilution, hemolysis, and decrease in serum electrolytes, except for potassium, by the transportation of hypotonic water into microvessels through type I alveolar epithelial cells; these changes, in particular hyperkalemia, often lead to critical heart arrhythmias. In forensic practice, distinguishing between FWD and SWD may critically help in the differentiation between unintentional/accidental and intentional (suicidal or homicidal) drowning. Thus, macroscopic, ultrastructural, and biochemical forensic studies on differentiation between FWD and SWD have been performed over the years [11,12,13,14,15,16,17].

In recent years new markers have been explored from the forensic perspective of both assessing drowning as cause of death and differentiating FWD and SWD, among which the typing of myelomonocytes [18], the evaluation of the surfactant protein A [19] and of aquaporins [20]. Aquaporins (AQPs) are small, integral membrane proteins (MW ~30,000) extensively distributed on the cell membrane that form pores and primarily act in several transporting and trafficking processes [21]. Forming pores at the level of biological membranes, AQPs act as selective channels allowing water transportation in many tissues, such as the kidneys, lungs, and brain [22]. Several AQPs have been investigated in the forensic practice in the differential diagnosis between FWD and SWD; in particular, the immune expression of intracerebral AQP4 [20], showing forensic reliability of the immunohistochemical analysis of intracerebral AQP4 expression for differentiation between FWD and SWD, and the intrarenal detection of AQP2 [23], both opening new diagnostic possibilities in the post-mortem differentiation between FWD and SWD. The immunohistochemical studies about intrapulmonary AQP5 expression in drowning have shown discordant results in differentiation between FWD and SWD [24,25,26].

Moving from studies showing the presence of AQP5 in the apical membrane of type-B intercalated cells (ICs) in the connecting tubule and cortical collecting duct of human kidneys [27], in this study, we investigated intrapulmonary and intrarenal AQP5 expression in forensic autopsy cases and discussed its suitability as a drowning marker and for differentiation between freshwater drowning and saltwater drowning.

## 2. Materials and Methods

**Case selection.** Cases were selected from the case series of the Section of Legal Medicine, University of Ferrara, Italy, and the Section of Legal Medicine, University of Foggia, Italy.

The study was conducted on tissues previously included in paraffin, and therefore should be considered retrospective. Samples were anonymized following the selection of cases based on cause and mode of death.

A total of 30 corpses diagnosed with drowning death, with 15 cases of freshwater drowning (FWD) and 15 cases of saltwater drowning (SWD) included. All cases were subjected to complete autopsy, standard histological examinations (hematoxylin–eosin) and diatom test. As controls (CTR), we selected a total of 17 subjects, who died from sudden cardiac death (*n* = 6), polytrauma (*n* = 4), overdose (*n* = 1), gunshot head injuries (*n* = 6).

In all cases SWD, FWD, and CTR were selected between male subjects, the age range was 20–50. The cases positive for alcohol and drugs on toxicological examination and with known diseases were discarded. A preliminary histological examination of brain, lung, heart, liver, kidney, and spleen was performed to select only healthy subjects.

**Technical details.** For all subjects included in the study, autopsy was performed between 24 and 72 h after death. The immersion intervals ranged from a few to 48 h. To avoid the progression of transformative phenomena, bodies were kept in a cold storage room (4 °C) until autopsy. Standard sample blocks were taken from lungs and kidneys. In each case, the tissue samples were fixed in 10% formalin and then processed and embedded in paraffin.

**Histological and immunohistochemical study.** For each case, lung and kidney sections of about 4 μm thickness were cut. The sections were immersed in a succession in trays containing xylene, two baths lasting 20 min each, then two 100% alcohol baths of 5 min each, then in 90% alcohol for 5 min, followed by a 20-min bath in 90% alcohol and hydrogen peroxide, then they were immersed in 70% alcohol for another 5 min and finally passed briefly in distilled water. Heat induced antigen retrieval (90 °C for 8 min) was performed using a 0.01 M pH 6 citric acid buffer. After cooling, the slides were washed with PBS. Aquaporin-5 Antibody NBP2-39043—Novus Biologicals was used as a primary antibody, diluted in PBS at 1:100 and incubated for two hours. The samples were subsequently developed with the CTS005 HRP-DAB system R&D kit, exploiting the avidin-biotin system, according to the manufacturer’s indications.

We evaluated the presence of non-specific markings due to the avidin-biotin system. Hence, we carried out tests using a polymer system (BioCare Goat-on-rodent HRP-Polymer), obtaining the markings of the same areas.

Sections were counterstained with hematoxylin, dehydrated, cover-slipped and observed in a Nikon Eclipse E600 microscope (Nikon, Tokyo, Japan). Quantification of AQP5 positive areas was performed using the ImageJ software (imagej.nih.gov/ij/, accessed on 11 July 2022) and expressed as an extension of the positive stained area.

**Statistical analysis.** Data were analyzed using GraphPad Prism 5 software for Windows (La Jolla, CA, USA). Data were checked for normality and analyzed by Kruskal–Wallis test, followed by Dunn’s multiple comparisons test. For all tests, a *p* value < 0.05 was considered statistically significant.

## 3. Results

### 3.1. Immunohistochemical Analysis of AQP5 in the Lung

First, we examined the distribution of AQP5 in the control lungs. As expected, AQP5 was expressed in pneumocytes and bronchial epithelial cells; AQP5-labeling positivity of intra-alveolar macrophages was observed too.

We observed a significant hypo-expression of AQP5 in cases of drowning in freshwater compared to those in saltwater and to controls (*p* < 0.0001). There was no statistically significant difference in expression between SWD and controls (Figure 1).

### 3.2. Immunohistochemical Analysis of AQP5 in the Kidney

We first observed the distribution of AQP5 in the renal control tissue. As reported [26], AQP5 was expressed in the cortical collecting duct system. Compared to the lung, a lower positivity was observed, both in terms of staining intensity and in the number and extent of positive areas.

Similar to what was observed in lung samples, we noticed a significant hypo-expression of AQP5 in cases of drowning in freshwater compared to those in saltwater and to controls (*p* < 0.0001). There was no statistically significant difference in expression between SWD and controls (Figure 2).

## 4. Discussion

The analysis of lung samples showed a clear hypo-expression of AQP5 in FWD cases compared to SWD cases and controls. This datum is consistent with the findings reported by Hayashi et al. [24] in a murine model and in human samples (by evaluating the mRNA coding for AQP5). This phenomenon can be explained by the electrolytic modifications induced by the inhalation of freshwater and the functionality of AQP5. In the case of FWD the drowning hypotonic medium induces the reabsorption of water from the alveoli to the bloodstream. AQP5 is localized in the apical membrane of type I pneumocytes, where it provides the major route for water movement across the alveolar epithelium [28]. A study on mice lacking AQP5 showed that these subjects exhibited a marked reduction of osmotic water permeability across the alveolar epithelium [29]. Similarly, it can be hypothesized that alveolar epithelium hypo-express AQP5 as a vital reaction to a hypo-osmotic insult.

Finally, no statistical difference was observed between SWD cases and controls. This evidence confirms what was observed by Hayashi et al. However, it appears to be in contrast with the results of in vitro studies, which showed hyper-expression of AQP5 in conditions of hyperosmolarity. This could be explained by the fact that in cases of hyperosmolarity the cell cycle, as well as RNA transcription, and protein synthesis are inhibited [30,31].

Renal expression of AQP5 in FWD cases was significantly lower than in SWD and controls (*p* < 0.0001). There is no direct contact of the renal collecting duct cells with the drowning medium, explaining how these cells modify their expression of AQP5 in the very fast time of FWD.

The authors who identified AQP5 in the human kidney [27] hypothesized that it could have an osmosensor role, given its reduced expression compared to other aquaporins (AQP2, AQP3) and the role of regulation that plays in other tissues (salivary glands) [32]. Wu H. et al. [33], investigating the functioning of the Dot1a gene, found that AQP5 protein was barely detectable in controls, but robustly expressed in mice with Dot1l deficiency in renal Aqp2-expressing cells (Dot1l(AC)), where it colocalizes with Aqp2. In the same experiment, using immunofluorescence on human kidney biopsies, it was found that in controls (patients with minimal change disease, MCD), AQP5 was not identifiable, while it was hyper-expressed in patients with diabetic nephropathy.

It is possible to hypothesize that in the kidney, with the appearance of hypo-osmotic plasma, AQP5 is hypo-expressed, as a vital reaction, to regulate the renal reabsorption of water.

In case of SWD drowning, a very small overexpression of renal AQP5 was observed, although not statistically significant. A possible hypothesis is that the SWD-induced hyperosmolarity is similar to the hyperosmolar state of patients with diabetic nephropathy, a condition that leads to an overexpression of AQP5.

## 5. Conclusions

The significant immunohistochemical hypo-expression of AQP5 in the samples of lungs and kidneys from FWD compared to those from SWD induces us to consider AQP5 as a reliable marker to which the pathologist may resort whenever dealing with the differential diagnosis between FWD and SWD [34,35]. The present study allows us to consider the immunohistochemical study of lung and renal tissues as a matter of paramount utility in the distinction between FWD and SWD, thus integrating the traditional forensic investigation [36]. The findings of the study can also be applicable to diagnose FWD, and not only to differentiating between FWD and SWD.

## Figures and Tables

**Figure 1 medicina-58-01458-f001:**
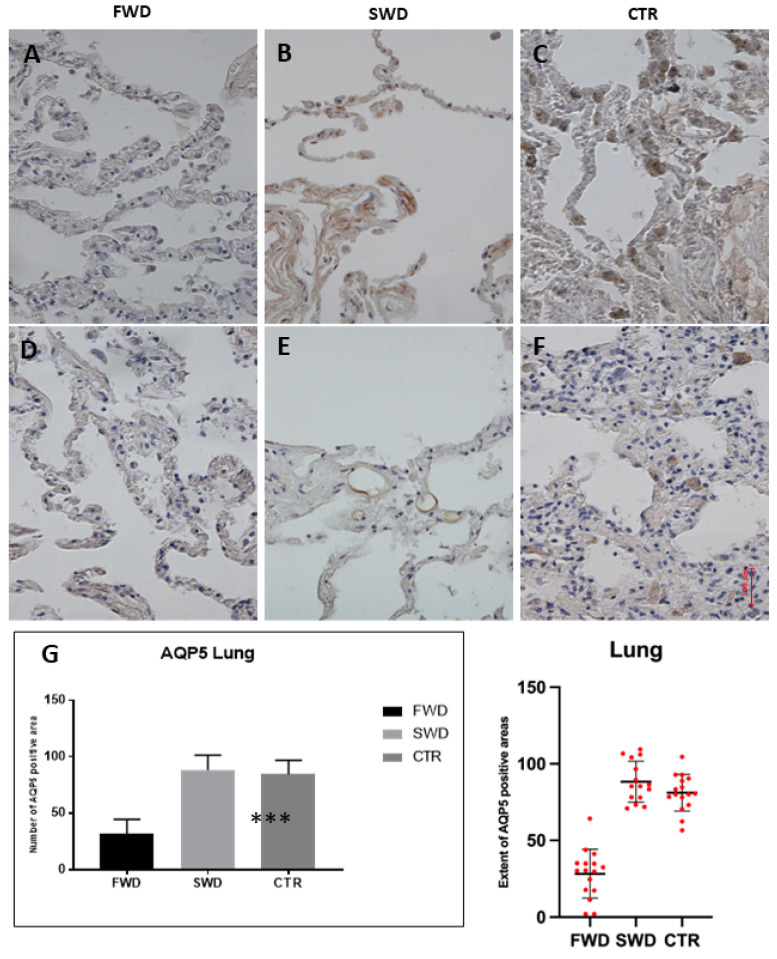
**Immunohistochemical reaction of AQP5 in the lungs**. (**A**,**D**): Freshwater drowning (FWD), weaker expression of aquaporin-5 (AQP5); (**B**,**E**): saltwater drowning (SWD); (**C**,**F**): control death due to brain gunshot wound (CTR); (**G**): AQP5 quantification. Data were checked for normality and analyzed by Kruskal–Wallis test, followed by Dunn’s multiple comparisons test. Significant hypo-expression of AQP5 in cases of FDW compared to those in SWD and to CTR *** *p* < 0.0001.

**Figure 2 medicina-58-01458-f002:**
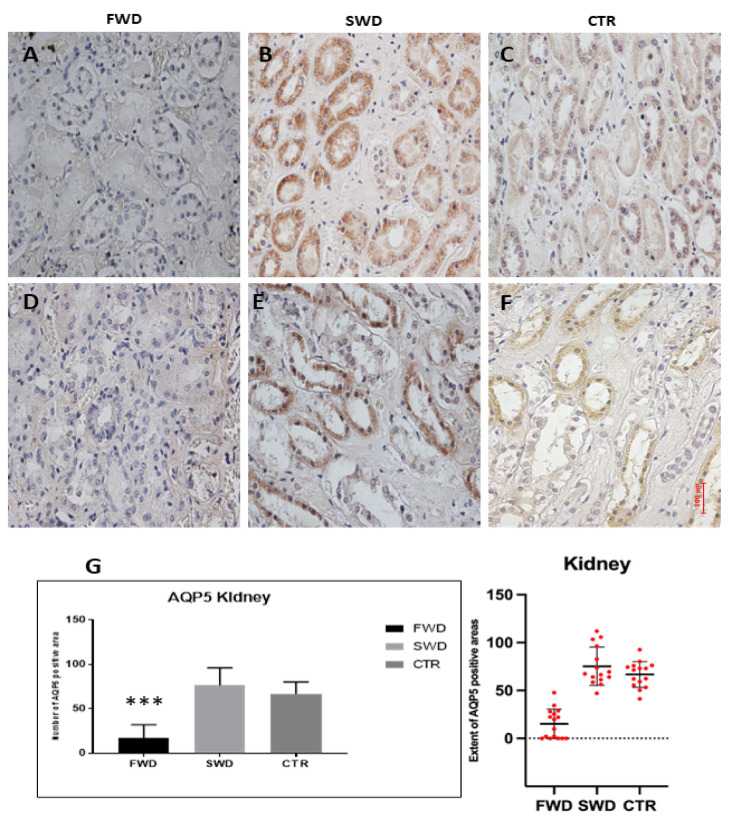
**Immunohistochemical reaction of AQP5 in the kidneys.** (**A**,**D**): Freshwater drowning (FWD), weaker expression of AQP5; (**B**,**E**): saltwater drowning (SWD); (**C**,**F**): control death due to brain gunshot wound (CTR); (**G**) AQP5 quantification. Data were checked for normality and analyzed by Kruskal–Wallis test, followed by Dunn’s multiple comparisons test. Significant hypo-expression of AQP5 in cases of FDW compared to those in SWD and to CTR *** *p* < 0.0001.

## Data Availability

The datasets generated during and/or analyzed during the current study are available from the corresponding author on reasonable request.

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
