# Peer review of "Forensic Diagnosis of Freshwater or Saltwater Drowning Using the Marker Aquaporin 5: An Immunohistochemical Study"

_medicina, 2022, doi:10.3390/medicina58101458_

Round 1
Reviewer 1 Report
The diagnosis of drowning and the differentiation between freshwater and saltwater drowning is always a challenge in forensic autopsies. The authors have presented the findings well. However, the results would be more obvious if the major statistical analyses were presented in tables.
Because of the low expression of AQP5 in freshwater drowning, I think the findings can also be applicable to diagnose drowning in freshwater, and not only to differentiate between freshwater and saltwater drowning. This can be added to the conclusion in the main text, not only in the conclusion of the abstract section.
Author Response
Reviewer1 (R1):
The diagnosis of drowning and the differentiation between freshwater and saltwater drowning is always a challenge in forensic autopsies. The authors have presented the findings well. However, the results would be more obvious if the major statistical analyses were presented in tables.
Because of the low expression of AQP5 in freshwater drowning, I think the findings can also be applicable to diagnose drowning in freshwater, and not only to differentiate between freshwater and saltwater drowning. This can be added to the conclusion in the main text, not only in the conclusion of the abstract section.
Thanks again for your appreciation of our study.
According to your suggestion and Editor indication, we added in figures 1 and 2 the statistical analyses presented in dot plots, the explain better the results.
We improved the conclusion as you suggest see lines215-216, the green highlight text.
A native speaker revise English of the text, corrections are highlighted in yellow.
Reviewer 2 Report
The reviewed manuscript “Forensic diagnosis of freshwater or saltwater drowning: an immunohistochemical study” presents a drowning marker aquaporin-5 (AQP5) to differentiate between freshwater drowning. Overall, the submitted manuscript is well organized and easy reading.But I do not find novelty aspects considering several works available in the literature. Below gives my comments and suggestions:
1. The contribution of the current work should be highlighted.
2. The novelty aspect of this work is not clear. I do not find novelty aspects considering several works available in the literature compared y Hayashi’s study except for the use of human specimens.
3. Some of the claims regarding “the analysis of renal and intrapulmonary AQP5 expression would be forensically useful for differentiation between FWD and death due to other causes ” The authors need to compare the results of all other death causes quantitatively in order to make such claims.
4. Line 80 “FSD” should be “FWD”
Author Response
Reviewer2 (R2): The reviewed manuscript“Forensic diagnosis of freshwater or saltwater drowning: an immunohistochemical study”presents a drowning marker aquaporin-5 (AQP5) to differentiate between freshwater drowning.Overall,the submittedmanuscript is well organized andeasy reading.ButI do not find novelty aspects considering several works available in the literature. Below gives my comments and suggestions:
1.The contribution of the current work should be highlighted.
2.The novelty aspect of this work is not clear. I do not find novelty aspects considering several works available in the literature compared y Hayashi’s study except for the use of human specimens.
3.Some of the claims regarding“the analysis of renal and intrapulmonary AQP5 expression would be forensically useful for differentiation between FWD and death due to other causes”The authors need to compare the results of all other death causes quantitatively in order to make such claims.
4.Line 80“FSD”should be“FWD”
Thanks again for your suggestions about our study.
“1. The contribution of the current work should be highlighted.
- The novelty aspect of this work is not clear. I do not find novelty aspects considering several works available in the literature compared y Hayashi’s study except for the use of human specimens.”
We are a team of forensic pathologists and in our Country, Italy is very important to use a validated method during Suite Court. Our paper aims to validate an immune histochemical marker of drowning in human samples. The excellent paper of Hayashi entitled "Differential diagnosis between freshwater drowning and saltwater drowning based on intrapulmonary aquaporin-5 expression" has been a source of inspiration for us. In our paper, we confirmed the results of Hayashi on lung human samples, and we improved the work including kidneys. The results of our paper from our point of view are very good because the positivity of Aquaporin 5 is a validated marker in the case of FWD in Court Cases.
“3. Some of the claims regarding “the analysis of renal and intrapulmonary AQP5 expression would be forensically useful for differentiation between FWD and death due to other causes” The authors need to compare the results of all other death causes quantitatively in order to make such claims.”
In the paper we compare three groups FWD, SWD and Control cases. Control cases are died from sudden cardiac death, polytrauma, overdose, gunshot head injuries, so it is an heterogenous kind of causes of death and the results are statistical significative. To better clarify the Control group, we improve in the text the information about the cases.
“4. Line 80 “FSD” should be “FWD”
We corrected in the text see line 89.
Reviewer 3 Report
Dear Authors
I read the article with great interest, and I think it can be published after the changes:
Firstly, in the introduction, it should be clearly stated that acute water lung emphysema is the most important and practically characteristic sign of drowning in freshwater;
Secondly, in cases of individuals who died by drowning in the freshwater (FWD), the volume of circulating blood increases causes hypervolemia, marked hemodilution, hemolysis, and decrease of serum electrolytes, except for potassium, by the transportation of hypotonic water into microvessels through type I alveolar epithelial cells. Finally, these changes (especially hyperkalemia) often lead to critical heart arrhythmias, so it cannot be said that both pathomechanisms of death by drowning are similar because it is not valid. These need improvement in the introduction;
Thirdly, I would like to know do fixing tissues in formalin is better than immediately frozen in liquid nitrogen and stored at −80°C until use;
Fourthly, the material and methods should have more information about the studied cases, such as age, sex, alcohol levels in blood and urine, and comorbidities. If the above changes have an impact or they may affect the obtained results, the limitations resulting from their presence should be described;
In the end, please also indicate, if possible, whether other diseases, disorders, or mechanisms of death will contribute to potential false positives, similar to those observed in the comparison of FWD to SWD.
Sincerely Reviewer
Author Response
Reviewer3 (R3):
I read the article with great interest, and I think it can be published after the changes:
Firstly, in the introduction, it should be clearly stated that acute water lung emphysema is the most important and practically characteristic sign of drowning in freshwater;
Secondly, in cases of individuals who died by drowning in the freshwater (FWD), the volume of circulating blood increases causes hypervolemia, marked hemodilution, hemolysis, and decrease of serum electrolytes, except for potassium, by the transportation of hypotonic water into microvessels through type I alveolar epithelial cells. Finally, these changes (especially hyperkalemia) often lead to critical heart arrhythmias, so it cannot be said that both pathomechanisms of death by drowning are similar because it is not valid. These need improvement in the introduction;
Thirdly, I would like to know do fixing tissues in formalin is better than immediately frozen in liquid nitrogen and stored at −80°C until use;
Fourthly, the material and methods should have more information about the studied cases, such as age, sex, alcohol levels in blood and urine, and comorbidities. If the above changes have an impact or they may affect the obtained results, the limitations resulting from their presence should be described;
In the end, please also indicate, if possible, whether other diseases, disorders, or mechanisms of death will contribute to potential false positives, similar to those observed in the comparison of FWD to SWD.
Thank you again for the very good recommendations about our paper.
“Firstly, in the introduction, it should be clearly stated that acute water lung emphysema is the most important and practically characteristic sign of drowning in freshwater;
Secondly, in cases of individuals who died by drowning in the freshwater (FWD), the volume of circulating blood increases causes hypervolemia, marked hemodilution, hemolysis, and decrease of serum electrolytes, except for potassium, by the transportation of hypotonic water into microvessels through type I alveolar epithelial cells. Finally, these changes (especially hyperkalemia) often lead to critical heart arrhythmias, so it cannot be said that both pathomechanisms of death by drowning are similar because it is not valid. These need improvement in the introduction;”
We provide to improve the introduction using your suggestions, see in the text the green highlighted modifications.
“Thirdly, I would like to know do fixing tissues in formalin is better than immediately frozen in liquid nitrogen and stored at −80°C until use;”
We chose fixing tissues in formalin because it is the routine method used in our forensic laboratories, our study is retrospective and we need homogeneous samples collecting, so we prefer a selection between the cases already performed. But we worked with frozen samples and it is a suitable method too.
“Fourthly, the material and methods should have more information about the studied cases, such as age, sex, alcohol levels in blood and urine, and comorbidities. If the above changes have an impact or they may affect the obtained results, the limitations resulting from their presence should be described;”
We select only healthy subjects and negative to the toxicological exams, we insert in the material and methods this information, see in the text the green highlighted modifications.
“In the end, please also indicate, if possible, whether other diseases, disorders, or mechanisms of death will contribute to potential false positives, similar to those observed in the comparison of FWD to SWD.”
In our samples we didn’t see false positive, just to be safe in control group we prefer a selection of various kinds of death, and we didn’t see a significative difference, the clear data is the hypo-expression of AQP5 in FWD.
Round 2
Reviewer 2 Report
The authors are unwilling or unable to address my concerns sufficiently to make this manuscript suitable for publication.